# Quanty-cFOS, a Novel ImageJ/Fiji Algorithm for Automated Counting of Immunoreactive Cells in Tissue Sections

**DOI:** 10.3390/cells12050704

**Published:** 2023-02-23

**Authors:** Carlo Antonio Beretta, Sheng Liu, Alina Stegemann, Zheng Gan, Lirong Wang, Linette Liqi Tan, Rohini Kuner

**Affiliations:** 1Pharmacology Institute, Heidelberg University, Im Neuenheimer Feld 366, 69120 Heidelberg, Germany; 2Department of Functional Neuroanatomy, Institute for Anatomy and Cell Biology, Heidelberg University, Im Neuenheimer Feld 307, 69120 Heidelberg, Germany

**Keywords:** quantitative analysis, immunohistochemistry, in situ hybridization, Fos protein, *c-fos* mRNA, 2D automated cell counts, open-source ImageJ/Fiji tool

## Abstract

Analysis of neural encoding and plasticity processes frequently relies on studying spatial patterns of activity-induced immediate early genes’ expression, such as *c-fos*. Quantitatively analyzing the numbers of cells expressing the Fos protein or *c-fos* mRNA is a major challenge owing to large human bias, subjectivity and variability in baseline and activity-induced expression. Here, we describe a novel open-source ImageJ/Fiji tool, called ‘Quanty-cFOS’, with an easy-to-use, streamlined pipeline for the automated or semi-automated counting of cells positive for the Fos protein and/or *c-fos* mRNA on images derived from tissue sections. The algorithms compute the intensity cutoff for positive cells on a user-specified number of images and apply this on all the images to process. This allows for the overcoming of variations in the data and the deriving of cell counts registered to specific brain areas in a highly time-efficient and reliable manner. We validated the tool using data from brain sections in response to somatosensory stimuli in a user-interactive manner. Here, we demonstrate the application of the tool in a step-by-step manner, with video tutorials, making it easy for novice users to implement. Quanty-cFOS facilitates a rapid, accurate and unbiased spatial mapping of neural activity and can also be easily extended to count other types of labelled cells.

## 1. Introduction

Analysis of neural circuits frequently relies on the use of immunohistochemistry assays to identify specific cell types using neurochemical marker proteins or mRNAs target genes. Similarly, quantitative analyses of cell counts expressing plasticity markers, such as the activity-induced immediate early gene, *c-fos*, represent a cornerstone of studying neural plasticity processes over large cellular networks in histological specimens. However, reliably counting cells immunohistochemically positive for protein markers or for mRNAs visualized in in situ hybridization experiments in large histological specimens, such as brain sections, remains a major challenge. Manual counting is extremely time-consuming, cumbersome and prone to subjective variations.

Recently, several automated digital image analysis tools have been developed. A major part of this development has focused on an automated analysis of the expression levels of proteins over regions of interest in histological specimens, such as tumor or immune markers, as clinical diagnostic or prognostic tools [1,2,3]. Despite this progress, availability of automated tools that enable easy-to-use, reproducible and reliable identification and quantitative counting of positive cells in immunohistochemical or mRNA in situ hybridization experiments on thick slices of tissue remains limited [4,5]. Often, user is required to have image analysis skills and experience, in-depth coding knowledge or access to expensive commercial software. Furthermore, automated tools for the identification of positive cells need high signal-to-noise levels, thus favoring highly expressed proteins. Signals representing nuanced differences in expression levels, high background and difficult antibodies, in contrast, are not suitable for conventional automated tools.

This is particularly relevant to the protooncogene *c-fos*, the immediate early gene that is directly induced in expression upon neuronal activation, leading to a rapid and transient build-up of *c-fos* mRNA and consequently of the Fos protein, which decays shortly after cessation of neuronal activity. Over the recent years, mapping Fos expression has emerged as a technically simple and reliable global marker for analyzing neurons that are activated by diverse external inputs, such as sensory stimuli [6]. Moreover, because Fos expression is well-correlated with behavioral readouts in animals, Fos-based mapping enables spatial analysis of regions and cells recruited during particular behaviors [6,7,8,9]. Finally, Fos expression is frequently used to characterize the effects of diverse therapeutic regimens on the central nervous system [10,11]. The recent development of Fos-based transgenic tools for labelling ensembles, i.e., cells that are co-active during particular functional tasks, as well as approaches that enable the consecutive labelling of two distinct cell populations with *c-fos* mRNA and the Fos protein, provide tremendous scope for studying functional encoding in the nervous system [7]. Thus, although they lack the spatial resolution of direct electrophysiological measurements, Fos-based mapping approaches represent attractive, highly useful and popular tools that are delivering unprecedented insights into neural function. Quantifying Fos-expressing neurons, however, still represents a major problem owing to high background levels and non-linear expression with different levels of signal-to-noise within and across samples preparations. Batch-to-batch variability of both antibody-based signals as well as *c-fos* mRNA in situ probes leads to several confounding effects. This has led to the necessary use of experimenter-based manual counting in studies employing Fos-based mapping in a quantitative manner, which is highly laborious, time-consuming, not entirely objective and highly subject to experimenter bias.

Here, we report the development of an open-source tool for ImageJ/Fiji [12], called ‘Quanty-cFOS’, with an easy-to-use, streamlined pipeline for automated or semi-automated quantitative analysis of cells positive for the Fos protein and/or *c-fos* mRNA on two-dimensional images (2D) or confocal maximum intensity projections (MIP) derived from tissue sections. Using example data sets of brain tissue from mice subjected to somatosensory stimuli, we demonstrate the entire process in a step-by-step manner and with the use of video tutorials, making it easy for novice users to apply on their images. Using manual counting to establish ground truth, we demonstrate both the fidelity of Quanty-cFOS and its ability to overcome user-to-user subjective variability. The tool also takes into account day-to-day and sample-to-sample variations in staining efficiency and enables for the deriving of cell counts registered to specific brain areas in a highly time-efficient and reliable manner. Thus, by delivering reliable and fast automated cell quantification across complex, technically non-optimal data sets, Quanty-cFOS accelerates the use of Fos mapping for analyses of neural circuits and thus provides an impetus to a wide range of research fields, including memory, chronic pain, addiction and psychiatric disorders. Importantly, although this tool was optimized and validated for quantitating Fos-expressing cells, it is just as readily applicable to any antibody, and is particularly suitable for proteins that show a variable baseline and induced expression within and across samples.

## 2. Materials and Methods

### 2.1. Animals

All experiments were conducted in C57BL/6J male mice (20–30 g) at 8 weeks of age that were obtained from Janvier Labs. In total 12 animals were used. Mice were housed individually in separated cages and kept under a 12 h light/dark cycle at a controlled temperature (22 ± 2 °C), humidity (40–50%), with food and water provided ad libitum according to ARRIVE guidelines. All experimental procedures were approved by the local governing body (Regierungspräsidium Karlsruhe, Germany, Ref. 35-9185.81/G-184/18), and abided to German Law (TierSchG, TierSchVersV) that regulates animal welfare and the protection of animals used for scientific purpose.

### 2.2. Application of Sensory Stimuli and Fos/c-fos Induction

A heat stimulus was presented to mice on a hot plate at 50 °C for 30 s (Ugo Basile Inc., Gemonio, Italy). Mice were exposed only once. To allow sufficient expression of Fos and to validate the Quanty-cFOS ImageJ/Fiji tool, C57BL/6J mice were kept in a home cage after stimuli for 20 min, 1 h or 3 h after the application of the stimulus prior to perfusion.

### 2.3. Tissue Fixation and Immunofluorescence Antibody Staining

Mice were sacrificed with an overdose of carbon dioxide, transcardially perfused with pre chilled phosphate-buffered saline (PBS) followed by 10% formalin fixative solution (Merck, Darmstadt, Germany). The brains were extracted and fixed in 10% formalin for 24 h at 4 °C. Coronal brain sections were collected at 50 μm with a vibratome (Leica VT100S, Wetzlar, Germany).

Free-floating sections were incubated in antigen retrieval solution (2.94% Tri-sodium citrate in dH_2_O, pH 8.5) for 25 min at 85 °C, and after cooling down, washed at room temperature with 50 mM Glycine (AppliChem, Darmstadt, Germany) for 10 min, followed by PBS for 5 min and 0.2% Triton X-100 (Carl Roth, Karlsruhe, Germany) in PBS (PBST) for 15 min. Lastly, sections were treated with 5% horse serum in PBST for 1 h, before incubating with the rabbit anti-Fos primary antibody (Ab190289, 1:1000 in 5% horse serum in PBST, Abcam, Cambridge, UK) at 4 °C overnight. The next day, sections were washed with 5% horse serum in PBST 3 times for 10 min and incubated with the donkey anti-rabbit Alexa 594 secondary antibody (Ab21206, dilution 1:700 in 10% NHS in PBS, Thermo Fischer Scientific, Waltham, MA, USA) for 2 h. After washing again with 5% horse serum in PBST 3 times for 10 min and in PBS for 10 min, Hoechst (#H367, 1:10,000 in PBS, Thermo Fischer Scientific, Waltham, MA, USA) was added for 10 min, followed by washing 3 times in PBST for 10 min each and in 10 mM TRIS-HCl for 10 min before mounting on glass slides with Mowiol.

### 2.4. c-fos mRNA In Situ Hybridization and Fos Immunofluorescence Co-Staining

For brain tissue preparation, mice were killed with CO_2_ at a defined time interval after the application of the external sensory stimulus and perfused with chilled PBS, followed by chilled 4% PFA. Brains were removed and held in 4% PFA for 3 h and transferred to 30% sucrose-PBS at 4 °C for 18–24 h. Brains were coronally sectioned with a cryotome (Leica CM1950, Wetzlar, Germany) at 50 µm and slices collected into 24-well plates with chilled PBS. All equipment was precleaned with RNaseZAP (Sigma RNaseZAP, Darmstadt, Germany), and all reagents were prepared using diethyl pyrocarbonate (DEPC)-treated PBS to avoid RNase contamination.

For mRNA in situ hybridization, the *c-fos* mRNA in situ probe was constructed according to the information on the Allen Brain Atlas website (http://www.brain-map.org, accessed on 15 February 2023). The RNA probe was generated via an in vitro transcription and labeled using the DIG-RNA-Labeling Mix kit and T7 RNA polymerase (Merck, Darmstadt, Germany), dissolved as a 1 µg/mL concentration in the hybridization solution (50% formamide (*v*/*v*), 5× SSC, 0.3 mg/mL Yeast tRNA and 0.5 mg/mL Salmon Sperm DNA). For *c-fos* tyramide-amplified in situ hybridization, slides were firstly washed 3-times with ice-cold PBS for 3 min and treated with acetylation buffer (0.25% acetic anhydride (*v*/*v*) in 0.1 M triethanolamine) for 10 min at room temperature. After rinsing once with cold PBS, cells were permeabilized with 0.3% TX100-PBS for 20 min at 4 °C. For in situ hybridization, tissues were pre-hybridized in hybridization buffer for 30 min. For hybridization, a *c-fos* anti-sense probe (diluted 1:200 in hybridization buffer) was applied and incubated overnight at 65 °C. The sense probe was applied to control slides. Post-hybridization, the tissue was washed twice with 2 × SSC/0.1% N-Lauroylsarcosine/50% formamide at 60 °C, rinsed in RNAse buffer (10 mM Tris, pH 8.0, 500 mM NaCl, 1 mM EDTA) and then digested with 20 μg/mL RNaseA in the RNase buffer for 30 min at 37 °C. This was followed by washing with 2 × SSC/0.1% N-Lauroylsarcosine and 0.2 × SSC/0.1% N-Lauroylsarcosine twice for 20 min at 37 °C and then rinsed once again with MABT (Maleic acid buffer with 1% of Tween 20). Tissue was then blocked with MABT++ (MABT with 10% heat-inactivated goat serum and 1% Blocking reagent) for 1 h at room temperature. Next, the tissue was incubated in MABT++ solution with the anti-digoxygenin antibody (anti-DIG-POD, 1:1000, Roche, Basel, Switzerland) at 4 °C for 16 h. For signal amplification, the slides were washed with MABT for 30 min at least 6 times, then rinsed with TSA buffer (10 mM imidazole) and incubated with TSA staining solution (Dilute Rhodamine tyramide 1:75 in TSA buffer, add in 0.001% H_2_O_2_) for 20 min at room temperature in the dark, followed by washing with PBST (PBS with 0.1% Tween 20) for 10 min, 5 times at room temperature in the dark. Tissues were mounted on slides with Mowiol after washing with PBS for single mRNA staining, or further used for immunofluorescence co-staining.

For immunofluorescence co-staining, the tissue was first washed with T-BST (0.05% Tween 20 and 0.05 M Tris-HCl in PBS) for 10 min, 5 times at room temperature in the dark and afterwards incubated with the anti-Fos primary antibody (1:1000, abcam, ab190289) in T-BST at 4 °C overnight. On the second day, the tissue was washed 3 times for 5 min in T-BST and then incubated with species-specific fluorescent secondary antibodies in T-BST for 1 h at room temperature. Finally, slides were washed 3 times for 15 min with 0.3% T-BST, then again with T-BST 3 times for 10 min and finally rinsed with 10 mM Tris-HCl for 10 min before mounting the coverslips with Mowiol.

### 2.5. Confocal Laser Scanning Microscopy Acquisition Settings

As examples of the region showing robust Fos expression upon somatosensory (cold) stimulation as well as reasonable baseline activity, coronal sections from the prelimbic and insula cortex (from 2.4 mm to 0.37 mm anterior to the bregma) were used for analysis. A confocal microscope (Leica TCS SP8, Wetzlar, Germany) was used to acquire immunofluorescence image stacks with 2 μm-thick planes using the 20× objective (N.A.: 0.75, oil immersion). Laser diode wavelengths of 405, 488, 552 and 638 nm in combination, respectively, with filters sets for DAPI (ex BP360/403 em LP425), FITC (ex 470/40, em LP515) and TRITC (ex 540/45, em LP590) were used. This resulted in an average z-optical section of 20 µm. The Fos protein signal showed nuclear staining pattern, whereas the *c-fos* mRNA appeared mostly in the cytoplasm.

### 2.6. Manual Counting of Cells Positive for Fos Protein or c-fos mRNA

For manual counting, the experimenter was blind to the different test groups, and images from groups were assigned a random number prior to analysis that was decoded after analysis. All confocal images were overlaid with the corresponding atlas section to anatomically define the regions of interest. All labeled cells within the boundaries of the defined sites were marked using a self-developed tool (https://github.com/cberri/cFOS_ManualAnnotations_ImageJ-Fiji, accessed on 15 February 2023) for ImageJ/Fiji and manually counted on MIP obtained from confocal stacks [12]. Brightness and contrast were optimized for each image. Background subtraction was performed by subtracting the mean intensity value estimated from a single background ROI placed within an unlabeled region in the same image. The Fos protein and *c-fos* mRNA signals were analyzed as separate images taken from the same slice using their respective excitation wavelengths. Positive cells on the XY boundary were excluded, and Fos protein signals were typically 6–8 µm in diameter and located in or near the nucleus. Nuclei were identified via DAPI staining. The *c-fos* mRNA signals were located in the cytoplasm as regions of 8–16 µm diameter surrounding the nucleus. In order to be counted positive, a cell had to display an intensity value above the intensity threshold of the background.

### 2.7. Quantification of Fos Protein and c-fos mRNA Intensity Features for Development of Quanty-cFOS

Image intensity features were extracted from 60 images acquired by 4 experimenters for the anti-Fos antibody staining and from 63 images acquired by 2 experimenters for the *c-fos* mRNA in situ using a customized ImageJ/Fiji script (https://github.com/cberri/Quanty-cFOS/blob/main/scripts/Extract-ImageProperties_V0.ijm, accessed on 15 February 2023). The script extracts intensity features to measure Fos staining variability between different preparations and image acquisition settings. The following features were computed for each image: mean intensity, standard deviation intensity, minimum intensity, maximum intensity, mode intensity and mean background intensity.

### 2.8. Development of Quanty-cFOS, an ImageJ/Fiji Tool to Count Fos/c-fos Positive Cells in an Unbiased Meter

Here, we developed *Quanty-cFOS.ijm* as an ImageJ/Fiji tool to semi-automatically or automatically count in an unbiased manner cells expressing the Fos protein or *c-fos* mRNA in fixed stained brain slices. It can be extended to generally count cell markers in 2D fluorescent images or on MIP. For flexibility reasons, this tool is developed as a macro-set (IJ1) for ImageJ/Fiji (tested on ImageJ version 1.53s and later) [12]. The proposed workflow consists of two major steps:-Automated cell segmentation,-Cell counting using the automated or the manual optimization method.

The Quanty-cFOS tool can be downloaded from https://github.com/cberri/Quanty-cFOS (accessed on 15 February 2023) and we provide a detailed step-by-step documentation of how to use it, including Appendix A and several scripts to validate the cell counting (the GitHub validation folder).

### 2.9. Automated Cell Detection with Quanty-cFOS

Quanty-cFOS cell detection is implemented using two different state-of-the-art segmentation strategies, based on deep learning and machine learning. The first uses the StarDist 2D *Versatile* (Fluorescent-Nuclei) inference available in the StarDist ImageJ/Fiji plugin and applies it on the raw images to segment convex shape structures [13]. The second uses the ilastik software pixel classification machine learning workflow to generate a probability map image using manual user annotations for different classes of pixels in an image [14]. In this case, the corresponding probability map image is loaded in addition to the raw input image in the Quanty-cFOS and intensity-thresholded to segment the cells. The user can decide which method is more suitable to process the images depending on the signal-to-noise ratio and on the shape of the cells that need to be segmented. The ilastik pixel classification workflow needs to be trained in ilastik software before to run the Quanty-cFOS tool (https://www.ilastik.org/documentation/pixelclassification/pixelclassification, accessed on 15 February 2023).

### 2.10. Automated Intensity Optimization Method in Quanty-cFOS

A key feature defining the novelty of the Quanty-cFOS counting method is the *z-score intensity cutoff* used for the *Automated Optimization*. The proposed algorithm computes the mean intensity value and the intensity standard deviation of each single segmented cell in the images, averages these two values and computes the z-score (*Zi*):$$Zi=\frac{xi-\mu }{\sigma }$$
xi: single cell intensityμ: mean cell intensityσ: mean cell intensity standard deviation

The intensity values in the significant z-score range (sigma) are averaged and used to set the intensity thresholds cutoff to count Fos/*c-fos*-positive (above) or negative (below) cells. The user can specify the range of standard deviations (sigma) to optimize the cutoff value for the Fos/*c-fos* cell counts. The larger the sigma value, the less restricted is the intensity cutoff value, and vice-versa.
$$IcS\;\left(Zi\to S\right)=\overset{n}{\underset{i=1}{\sum }}i\frac{xi}{n}$$
IcS: intensity cutoff computed on an imageZi: z-scoreS: significant range of standard deviations (sigma)xi: single cell intensityn: number of positive cells in an image


The cutoff optimization is critical to gain an accurate and robust estimation of cell numbers. To consistently calculate the mean and standard deviation intensity, an arbitrary number of images can be used as input to compute the intensity cutoff (see *Batch Analysis* with *Optimization Steps*). In this case, the average intensity values of the images used for the optimization are accounted to calculate the intensity cutoff.
$$IcA=\overset{f}{\underset{i=1}{\sum }}i\frac{IcS}{f}$$
IcA: intensity cutoff with the optimization stepsIcS: intensity cutoff x imagef: optimization steps


The results of the automated intensity method can be validated by manually counting cells in fewer images and by running the MATLAB correlation analysis provided together with the Quanty-cFOS (CorrelationAnalysis.mlx, see also the ValidationTable.xlsx file as an example). Manual counting can be performed using any favorite tool or by using the following ImageJ/Fiji IJ1 script that we developed for this purpose (https://github.com/cberri/cFOS_ManualAnnotations_ImageJ-Fiji, accessed on 15 February 2023).

### 2.11. Manual Intensity Optimization Method in Quanty-cFOS

The intensity threshold value used for Fos/*c-fos* cell counting is the key parameter to decide the cutoff for positive or negative counts. This is rather important if images have been acquired with different settings or high staining variability occurs between samples. To optimize this process and to help test different threshold values in a semiautomated unbiased way, we implemented the *Manual Optimization* function. By using this method, images can be previewed, and different intensity values can be tested for Fos/c-*fos*-positive cell counting. The *Manual Optimization* default intensity value displayed in the user setting dialog box is computed via the *Automated Optimization* function to help in choosing the appropriate intensity cutoff value. Moreover, different size filters for the cell area can be applied to remove small and large detected objects in the images. The number of images previewed is specified using the *Optimization Steps*. Indeed, only these images are used for testing different thresholds and the average intensity value of these thresholds is applied as an intensity cutoff on all the subsequently listed raw images.

### 2.12. Cell Batch Analysis without Intensity Optimization in Quanty-cFOS (Optional)

Counting all cells without intensity optimization is also possible as an option and can be achieved by unchecking the *Automated Optimization* and the *Manual Optimization*. In this way, all the cells in the image are counted without an intensity cutoff. Only a size cutoff filter (based on the cell area) is applied to exclude cells below the cutoff value and 5 times above the specified cutoff value. This option is supported only with *Batch Analysis* and the number of *Optimization Steps* is ignored.

### 2.13. Additional Quanty-cFOS Functionalities

Select *Multiple Sub-Brain Region* was added to select specific regions of interest in the input images and to count positive cells only inside the selected regions. This option works only without batch analysis.

Select *Allow Preview User Setting* was added to preview the intensity threshold and area cutoff used for the ilastik probability map segmentation (simple method). The intensity threshold method and cell size filter (area) can be modified to gain the best segmentation results. Currently, we support simple ImageJ/Fiji thresholding methods to segment cells in the ilastik probability map.

### 2.14. Additional Software

Matlab (R2019a) was used for the correlation analysis and statistics. Figures were prepared using Adobe Photoshop CS6 (Adobe) and Adobe Illustrator CS6 (Adobe). DaVinci Resolve was used to edit the Appendix A.

### 2.15. Statistical Analyses

All statistical analyses were performed in Matlab (R2019a). Box plots were created using the Matlab box plot function and show the mean intensity value +/− and the standard deviation (S.D.) of each plotted feature. The black horizontal line in each box represents the median value z-score. Analysis was computed in Matlab, and a positive correlation was considered in the range of two standard deviations. Box plots were generated for each time point for *c-fos* mRNA and Fos protein counts. Each box shows the mean intensity counts, the vertical lines show the S.D. (+/−).

## 3. Results

### 3.1. Fos/c-fos Staining Can Lead to Biased Results Depending on Sample Preparation and Microscopy Acquisition Settings

Our past experience has shown that quantifying Fos-expressing cells is challenging, not only because major differences exist in expression levels across cells as well as across samples, but also owing to technical aspects of sample preparation and imaging parameters. This was again observed when we acquired Confocal Laser Scanning Microscopy (CLSM) z-stacks after Fos protein immunostaining and *c-fos* mRNA in situ hybridization, as described under methods. To address differences in the image acquisition, four different experimenters prepared the samples, optimized the confocal settings and acquired the images (Figure 1A–D,A’–D’). We quantified Fos protein expression by extracting intensity features along the different staining and acquisition settings (ImageJ/Fiji Set Measurement plugin) [12]. Our analysis revealed major differences between the different extracted features across samples and experimenters. Indeed, we observed a high variability for the mean image intensity, mode intensity and mean background intensity features (Figure 1E). Intriguingly, the maximum intensity feature also showed a large dynamic range, suggesting fluctuation in the signal-to-noise ratio between acquired images. Moreover, for the minimum and the mode intensity, several data points were detected outside the whiskers in box-and-whisker plots, highlighting the differences between stained images (Figure 1E, + symbol).

The *c-fos* mRNA was evaluated in the same way on two different sets of samples prepared by two experimenters. The analysis revealed an even larger variability in terms of the mean image intensity, mode intensity and mean background intensity features (Figure 1E’). Differently from the Fos protein, the maximum intensity value is set to 255 for an 8-bit dynamic range (0–255), indicating that all the images have been saturated while being acquired. This is indicative of a low signal-to-noise ratio for the *c-fos* mRNA that required a high confocal gain or/and laser power during image acquisition. Considering the extracted intensity variability between the tested images, fewer data points were detected outside the whiskers (Figure 1E, plus symbol). Intensity fluctuations between different staining rounds, acquisition settings and in between images acquired to investigate a specific physiological problem can lead to bias, especially if cell counting is the main readout. These experiments thus demonstrate the need to reduce bias in manual counts, which served as the starting point of our efforts toward developing the Quanty-cFOS ImageJ/Fiji tool for the automated/semiautomated counting of cells positive for the Fos protein and *c-fos* mRNA.

### 3.2. Fos/c-fos Cell Counting Workflow with Quanty-cFOS

The Quanty-cFOS was developed to be a user-friendly, unbiased ImageJ/Fiji tool for Fos protein and *c-fos* mRNA counts. The workflow consists of four major steps: input, detection, quantification and results (Figure 2, Fos protein and *c-fos* mRNA workflow). An input directory containing all the Fos protein images to process can be chosen. For the Fos protein detection, we used the StarDist 2D versatile fluorescent nuclei model in the Quanty-cFOS (Figure 2A and Appendix A) [13]. This method generated labeled images (Figure 2B) and it was optimized to segment convex objects in two dimensions (2D). The cell detection can be easily improved in the Quanty-cFOS tool by training a custom StarDist 2D model [15]. For the *c-fos* mRNA counts, we used two input directories. The detection method uses pre-processed images obtained from ilastik pixel classification workflow [14] in combination with the raw images (Figure 2A1′,A2′; Appendix A). We implemented this method to detect any cell shape, from convex to more elongated shapes. Moreover, this option allows us to choose any pre-processed method in case cell detection is inefficient and upload the pre-processed images in the tool. MIP are automatically created using the Quanty-cFOS tool or can be generated by the user prior to the cell counting. The Quanty-cFOS tool supports three methods for cell counting: ‘*automated optimization*’, ‘*manual optimization*’ and ‘*all cells counts*’–‘*with batch analysis*’ and ‘*without batch analysis*’ (Appendix A; Appendix A). The option ‘*with batch analysis*’ has been implemented to help the user in choosing the detection parameters and the intensity cutoff to batch process all the images in the input source directory by applying the same settings. This modality allows us to apply the ‘*automated optimization*’, the ‘*manual optimization*’ and ‘*all cells counting*’ methods (Appendix A left; Appendix A).

Choosing the detection parameters can be difficult, in particular when the images are different from each other, or the counting method settings needs to be changed during the processing. Therefore, to simplify cell counting, we developed the option ‘*without batch analysis*’ (Appendix A right; Appendix A). Optimization methods and parameters can be changed for each image to achieve the best counts. This method is recommended if the images to process are very different from each other, or different parameters need to be tested for the cell counting. Chosen parameters are thereby saved in the root output directory to document the analysis (output Log.txt file). This modality can be used only for the ‘*automated*’ and the ‘*manual optimization*’ methods. Moreover, the option ‘*without batch analysis*’ supports the multiple ‘*sub-brain regions selection*’ function to count Fos-positive cells in selected subregions of an image (Appendix A, Appendix A).

The ‘*automated optimization*’ method is used to compute the intensity threshold cutoff on a predefined number of images specified via the ‘*optimization steps*’ and applies this cutoff to all the following images listed in the input directory (Appendix A, Appendix A). The cutoff intensity threshold is computed on the optimization images by calculating the z-score. Only cell intensity values in the range of the specified z-score sigma (number of standard deviations) are averaged and contribute to the final intensity cutoff. The ‘*manual optimization*’ method allows the user to choose the intensity cutoff by previewing a selected number of images specified via the ‘*optimization steps*’. These values are averaged and used as an intensity cutoff for cell counting (Appendix A; Appendix A). For both methods, an area filter is applied. For the ‘*automated optimization option’,* the area is set two times above and below the area standard deviation. In the ‘*manual optimization*’, the area can be measured on the previewed images and the cutoff value can be specified. The option ‘*All cells counts*’ can be used to count all the positive cells in the images without any ‘*optimization steps*’ (Appendix A, Appendix A). This has been included to count in 2D all the cells in an image independently of an intensity cutoff.

To simplify further analysis and statistics, the Quanty-cFOS output consists of an output root directory created outside the input path with subdirectories named as the input processed images. Each subdirectory saves the labeled image for positive and negative cells (Figure 2C,C’), the ImageJ/Fiji ROI Manager ROIs and a comma separator values (csv) file with the coordinates of the center of mass of each detected cell. Moreover, the output root directory contains the summary of the counts as an csv file and the Log.txt file with the analysis steps and the chosen parameters (Figure 2 and Appendix A). An additional subdirectory with all the labeled images for positive and negative cells is created in the main output path for further analysis.

### 3.3. Quanty-cFOS Validation for Fos Protein Cell Counting

After establishing the methodology, cell counting results generated using Quanty-cFOS on cells expressing the Fos protein were compared to manual counts of Fos-expressing cells using a test data set of randomly selected images from mouse brain sections (Figure 3). Fos-expressing cells were manually annotated in the validation images by four different experimenters (Figure 3A–D) and the results compared with the Quanty-cFOS output (Figure 3E). For the manual counting, we developed an ImageJ/Fiji tool that allows the user to select positive and negative cells by clicking the left and right mouse button. Positive cells can be counted by clicking the left mouse button, negative cells with the right mouse button (https://github.com/cberri/cFOS_ManualAnnotations_ImageJ-Fiji, accessed on 15 February 2023). By comparing the manual counts from the four experimenters, we observed consistent results between the single human counts in certain ROIs but also miscounted cells in other parts of the images (Figure 3A’–D’). Indeed, the single counts analysis of Fos protein-expressing cells over 30 images showed a discrepancy in the absolute number of positive cells counted manually (Figure 3F, Manual Count: experimenter one, experimenter two, experimenter three, experimenter four). The discrepancy in counts was also seen when the absolute single counts from manual counting were compared with the Quanty-cFOS output (Figure 3E’,F, automated Quanty-cFOS counts). To evaluate the relation between the two methods, we compared the manual counts average slope with the Quanty-cFOS slope. The slope for both, the manual average and Quanty-cFOS, showed a similar distribution in the number of positive cells counted, suggesting a consistent relation between the manual and the Quanty-cFOS counts (Figure 3G). To further verify this observation, we computed the correlation analysis between the manual and Quanty-cFOS counts using the z-score method. The analysis showed a significant correlation in two standard deviations range (sigma) between the manual and the Quanty-cFOS counts (Figure 3H). These results support the accuracy of the automated Quanty-cFOS method. Furthermore, the large differences in the absolute manual counts between the four experimenters further revealed the need of an unbiased algorithm for cell counting.

### 3.4. Quanty-cFOS Validation for c-fos mRNA Cell Counting

In Quanty-cFOS, we developed a similar approach for counting cells positive for *c-fos* mRNA in in situ hybridization experiments, as was used for the Fos protein. First, we compared the manual counting results performed by four experimenters on 30 images from the mouse brain with the output of the automated Quanty-cFOS algorithm (Figure 4A–E). The manual counts were obtained as described for the Fos protein. The automated counts were achieved by using the combination of raw and ilastik pixel classification probability map as pre-processing step. Ilastik pixel classification workflow was trained using 15 raw *c-fos* mRNA images and all the images were batch processed in ilastik [14]. A larger amount of training data or different pre-processing methods can be used to gain higher accuracy in the segmentation results, e.g., ilastik Autocontex [16], image denoising with noise2void [17], image restoration with CARE [18] and a suitable deep learning model from BioImage Model Zoo [19]. The manual counts comparison showed consistent results between the experimenters’ counts in certain ROIs but also revealed miscounted cells in other parts of the images (Figure 4A’–D’). Similar results are observed for the automated *c-fos* mRNA counts (Figure 4A’–E’). Manual cell counts showed differences in the absolute number of *c-fos*-positive cell counts (Figure 4F: manual counts: H1, H2, H3, H4). This was also seen when comparing the *c-fos* manual counts with the automated counting results (Figure 4F: automated Quanty-cFOS counts). However, the discrepancy within the absolute counts, within the manual counts and in between the manual and the automated counts was smaller in comparison to what we observed for Fos protein counts (Figure 3F). We evaluated the counts relation by comparing the manual counts average slope with the mRNA Quanty-cFOS slope. Both slopes, manual and automated, showed the same distribution with many overlapping data points (Figure 4G). We further tested the counts for significance by computing the z-scores in two S.D. ranges. Manual and automated counts showed a strong correlation. As with the Fos protein counts, these analyses demonstrate the validity and accuracy of the Quanty-cFOS automated method for counting *c-fos* mRNA-positive cells in complex tissues (Figure 4H).

### 3.5. Hot Plate Analysis of Fos Protein and c-fos mRNA Expression Using Quanty-cFOS

Having established the Quanty-cFOS automated method, we then demonstrate its utility for studying activation in neuronal networks in the rodent brain by applying it to follow changes in both *c-fos* mRNA and Fos protein expression after sensory stimulation. In mice subjected to a heat stimulus of 50 °C applied to the hindpaw, *c-fos* mRNA levels started to increase in the prefrontal cortex within 20 min, peaked at 1 h and the mRNA was degraded by 3 h after stimulation (Figure 5A–C). In contrast, Fos protein increased only after 1 h and strong expression was evident 3 h after stimulation (Figure 5D–F). The automated counts were compared with manual counts performed by four experimenters, as described above. Both automated and manual counts for mRNA and protein show the expected *c-fos* expression. Indeed, *c-fos* mRNA could be seen at 20 min, reached a peak after 1 h and was degraded after 3 h, while protein expression was evident at 1 h and not seen at 20 min post-stimulation (Figure 5G, blue and red dash lines). Thus, we validated the Quanty-cFOS method, showing that it can be reliably used to automate the Fos protein and *c-fos* mRNA cell counts. This is an important prerequisite for using Quanty-cFOS for the automated quantification of positive cell counts in methods involving dual counting of mRNA and protein within the same specimen. While this can apply to any biological marker mRNAs or proteins, the ability to reliably count cells expressing *c-fos* mRNA and protein within the same specimen in the context of the different time course of their expression using Quanty-cFOS will allow for the implementation of this tool in dual-epoch labelling methods, such as TAI-FISH, that involve an analysis of cells responding in an activity-dependent manner to different stimuli (e.g., two distinct sensory stimuli, such as heat and cold) applied with a temporal gap [7]. The variability of manual cell counts can affect the final experiment outcome depending on how an experimenter visually counts cells (Figure 5G, orange and green arrowheads). Instead, the Quantity-cFOS tool can be used to obtain unbiased cell counts making analysis reproducible and objective, as shown in our results.

## 4. Discussion

This study introduces an open-source, novel and fully validated ImageJ/Fiji tool for the unbiased counting of cells expressing Fos protein or c-fos mRNA. The main advantages of this tool are its objectivity and lack of human bias in cell counting, consistence of methodology and analyses across different experiments and its ability to set thresholds objectively in experiments with inter- and intra-experiment variability. Further, Quanty-cFOS allows for higher speed and efficiency in analyzing a large number of images and applicability to antibodies or RNA probes that yield high experimental variability as well as graded nuances in expression levels which can lead to large errors when viewed subjectively. Importantly, the study provides an easy-to-use tool, including in-depth step-by-step videos for different cell counting applications that can be quickly learned and efficiently applied by non-experts in image analysis.

Analysis of the activity-induced expression of the immediate early gene *c-fos* has rapidly established itself into a major surrogate for addressing activity in neurons [11]. While levels of expression of the gene can be measured quantitatively in terms of quantitative PCR analyses for the mRNA transcript or Western blot analyses for the protein product, they lack spatial and cellular resolution. Therefore, immunohistochemistry and in situ hybridization of the *c-fos* gene expression is critical in yielding information on activity-induced changes in distinct regions, pathways and individual cells of the brain. Although it lacks the fine temporal resolution of electrophysiology, analysis of spatial profiling of Fos expression is technically much easier, faster and can be carried out simultaneously across the whole brain, making it a broadly applicable method. Cells that induce and express Fos simultaneously following a particular external stimulus, such as painful sensory stimuli, or in association with a particular internal function, such as establishment or recall of particular memories, have been considered to be part of assemblies or ensembles that subserve particular functions. A very recent study has employed simultaneous Fos monitoring and in vivo calcium imaging of the hippocampus in mice to demonstrate that neurons with high Fos induction form cellular ensembles which show highly correlated activity and play an important functional role in spatial memory, thus leading further credence to the use of Fos mapping for identifying functionally relevant cells [20]. Thus, spatial profiling of the Fos protein is a valuable and widely employed method in the neurosciences. Unfortunately, quantitative analysis of Fos-expressing neurons has not evolved at the same rate, and most studies have relied on manual counting, which is not only highly cumbersome and time-consuming, but also prone to subjectivity, bias and variability within and across groups and experiments. While an automated algorithm was successfully developed for counting Fos-expressing cells in light sheet microscopy on cleared whole brains [21], analysis of thick brain slices or sections has proven to be more complex, owing to the high background from the brain parenchyma as well as difficulties in estimating cells that are cut out of the section in the z dimension. Fewer, automated solutions for cell counting on tissue sections are available as open-source standing alone software or as ImageJ/Fiji plugins. For instance, Cellpose and CellProfiler can be used to automatically count the number of cells in microscopy sections; however, these tools lacked specific thresholding optimization methods essential for mapping immunoreactive cells [4,5]. Basal expression of Fos/*c-fos* in neurons requires the development of ad hoc optimization algorithms necessary to count only reactive cells. Therefore, to cover this gap, we developed the Quanty-cFOS tool, optimized to count cells above an automated computed intensity or manual intensity cutoff. The option to count only cells above a specific cutoff, in a step-by-step designed workflow, is the major novelty of the Quanty-cFOS tool in comparison to the tools publicly available. To the best of our knowledge, there is no other open-source tool that is easily accessible and usable by non-experts, can run in a standard image analysis program such as ImageJ/Fiji, is accompanied by detailed video tutorials and which has been validated and optimized for both the Fos protein and c-fos mRNA. Currently, Quanty-cFOS can be used to count cells only on two-dimensional images and this could limit its application if cells need to be counted in three-dimensional stacks. Indeed, extending the Quanty-cFOS to count cells in three-dimensional stacks will be part of the future development of this open-source tool.

Here, we took care to include human experimenters and manual counting in an iterative process while developing and optimizing the Quanty-cFOS tool. Moreover, emphasis was placed on collecting data sets showing a high level of inter-experimenter variability to challenge and optimize Quanty-cFOS. Furthermore, an important challenge in counting Fos-positive cells in response to a given stimulus or function is given by the fact that natural, spontaneous activity yields background Fos expression and stimulus-derived Fos expression can vary in strength across cells within the sample and across samples, rendering it difficult to set a threshold. This aspect is dealt efficiently in Quanty-cFOS using the automated and manual optimization methods. Nevertheless, the experimenter can choose to either employ the automated optimization method, which represents the most unbiased option, or to use the manual optimization method, or to count all stained cells without any intensity cutoff, as required per experimental conditions, thus providing maximum flexibility.

Although in situ mRNA hybridization represents a more cumbersome method of spatially testing activity-induced expression of *c-fos*, requiring stringent control of RNA degradation, methods such as RNAscope have led to widespread use in recent times. Counting *c-fos*-positive cells harbors the same difficulties as discussed above for the Fos protein while carrying the additional hindrance that mRNA is diffusely distributed in a spotty, dotted appearance across the cytoplasm in contrast to nuclear localization of the Fos protein, rendering it harder to distinguish between neighboring cells. In Quanty-cFOS, using image preprocessing with ilastik pixel classification enabled us to circumvent this problem and facilitate the segmentation and the counting of mRNA-positive cells. This will not only foster the use of the *c-fos* mRNA as a spatial marker of activity-induced changes in networks, but also support methodologies in which analyses of the *c-fos* mRNA and Fos protein are combined with the same samples to identify differentially-activated cohorts, such as TaiFISH [7]. This method applies sequentially given stimuli and it takes advantage of the early expression and short lifespan of the *c-fos* mRNA in comparison with the later induction and longer expression of the Fos protein. Methods such as these are becoming increasingly prominent in yielding insights into the differential cellular encoding of distinct functions within the same region in the nervous system, e.g., aversion vs. reward processing in the prefrontal networks.

Furthermore, it deserves to be noted that emphasis was placed on making Quanty-cFOS user-friendly by developing several workflow options and leaving control in the hands of the user. Additionally, with the tool we provide in-depth video tutorials to make it easily useable by scientists without expertise in image processing and image analysis. This is of critical importance since the lack of bioimage analysis and computing skills often limits stringent standards fundamental for reproducibility in image quantification.

Finally, the efficiency of using Quanty-cFOS to quantitate activity-dependent changes in cellular responses deserves to be discussed. Having conducted and published a number of in-depth studies with c-fos-based activity mapping using manual counting that required several weeks to months of work [8,9], frequently leading to experimenter fatigue, we are confident that automated counting via Quanty-cFOS can achieve the same goals in the fraction of time required for manual analyses.

In conclusion, making this fully validated tool freely available to the scientific community will help overcome human bias in spatial activity mapping and foster unbiased, efficient and rapid analyses. Moreover, although the tool was designed and optimized for quantitating cells expressing the Fos protein or *c-fos* mRNA in the nervous system, it is in principle applicable to any antibody or mRNA being investigated and to any type of tissue, thus rendering its multiple applications.

## Figures and Tables

**Figure 1 cells-12-00704-f001:**
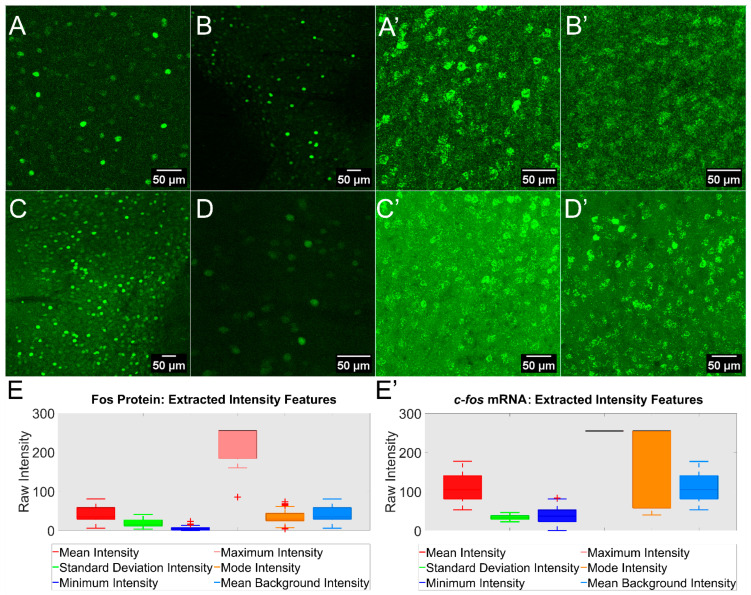
Intensity variability in Fos protein immunohistochemistry and mRNA in situ hybridization. (**A**–**D**) Confocal maximum intensity projections of neurons immunostained for expression of Fos protein in mouse brain sections (prefrontal cortex shown here), which were acquired by four experimenters. The representative images show staining variability and different confocal settings for image acquisition (images were acquired at different resolutions). (**A’**–**D’**) Confocal maximum intensity projections of c-fos mRNA after in situ hybridization on mouse brain sections, which were acquired by two experimenters (prefrontal cortex and S1 shown here). Representative images show differences in mRNA in situ hybridization efficiency and image acquisition settings. Scale bar was added using ImageJ/Fiji on the right bottom corner of each image. (**E**,**E’**) Intensity features box plot comparisons, in which intensity features were extracted using the Extract-ImageProperties_V0.ijm ImageJ/Fiji script. The extracted intensity features are shown plotted on the x-axis. Data points outside the whiskers are marked with the plus symbol. Data are represented as mean ± standard deviation (S.D.).

**Figure 2 cells-12-00704-f002:**
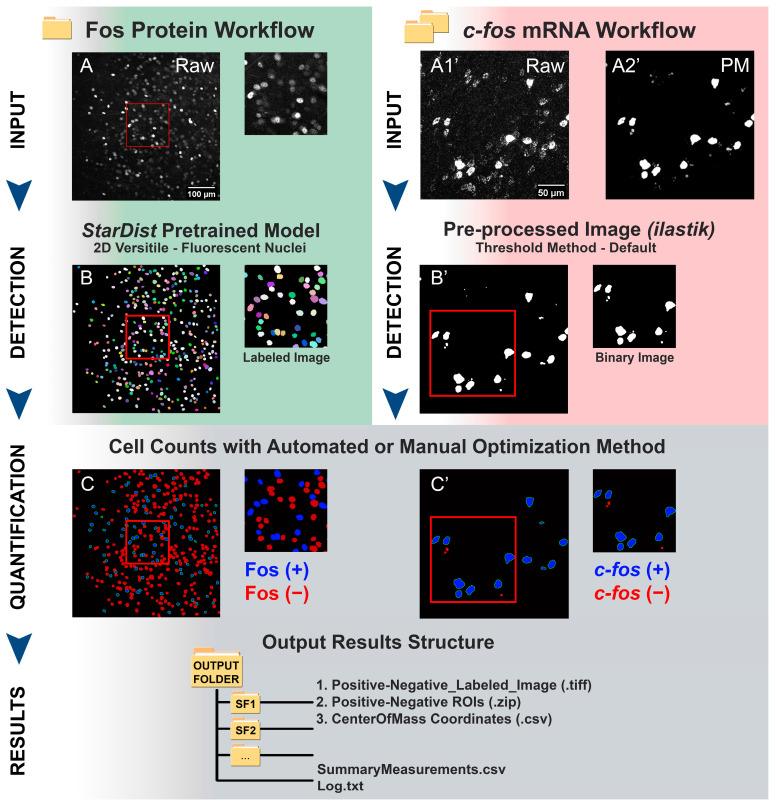
Step-by-step counting of cells expressing Fos protein and *c-fos* mRNA using the Quanty-cFOS ImageJ/Fiji tool. Top to bottom, Quanty-cFOS workflow steps: number of input directories containing the images to process (INPUT), detection method for image segmentation (DETECTION), positive cell counting (QUANTIFICATION, automated or manual optimization method) and output results (RESULTS). (**A**–**C**) Quanty-cFOS workflow to count Fos protein-positive neurons in confocal maximum intensity projections. (**A**) Representative input raw image, (**B**) StarDist segmented labeled image and (**C**) image showing output of positive and negative cells counted in Quanty-cFOS. Fos-positive cells (Fos+) are labeled in blue and Fos-negative cells (Fos-) are labeled in red. (**A′**–**C′**) Quanty-cFOS workflow to count *c-fos* mRNA-positive neurons in confocal maximum intensity projection images. (**A1′**) raw input image, (**A2′**) pre-processed ilastik pixel classification probability map input image. (**B′**) Thresholded ilastik pixel classification binary image (0, 255) generated via the Quanty-cFOS tool using the ImageJ/Fiji *default* threshold method. (**C′**) *c-fos* mRNA-positive cells (*c-fos+*) are labeled in blue and *c-fos* negative cells (*c-fos-*) are labeled in red. Red squares show a region of interest magnified on the right side of each image for Fos protein and *c-fos* mRNA. The output results folder structure is shown at the bottom (SF1, subfolder 1). (**A**,**A1′**,**A2′**) Scale bar was added using ImageJ/Fiji on the right bottom corner of each image.

**Figure 3 cells-12-00704-f003:**
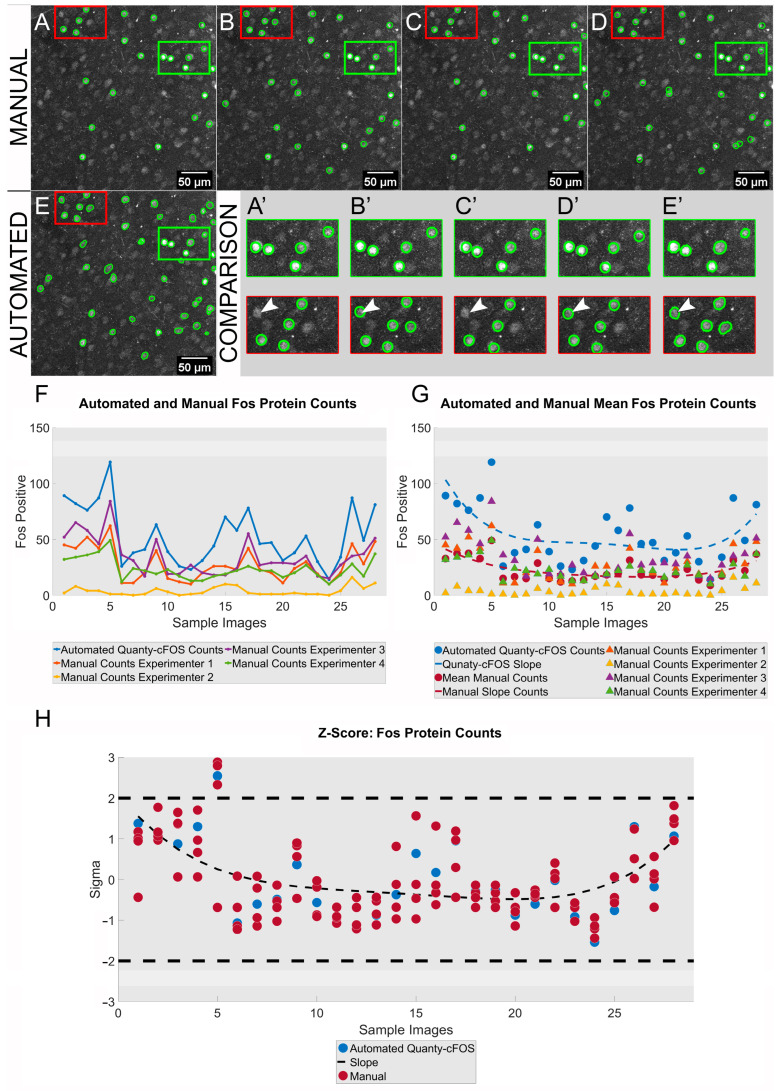
Fos protein: analysis and comparison of cells counted manually vs. Quanty-cFOS. (**A**–**D**) Representative maximum intensity projection images for Fos-positive cells manually counted by four experimenters. (**A**–**E**) Scale bar was added using ImageJ/Fiji on the right bottom corner of each image. (**E**) Quanty-cFOS automated Fos-positive cell counting using the ‘*Automated Optimization*’ method. (**A**–**E**) Fos-positive cells are highlighted with a green outline. (**A**–**E**,**A’**–**E’**) Green rectangular ROI shows cells counted manually by the 4 experimenters to be positive for Fos or counted as positive using the Quanty-cFOS automated method. Red rectangular ROI shows Fos-positive cells counted differentially between the four experimenters and the Quanty-cFOS tool. (**A’**–**E’**) Highlighted red and green rectangular ROIs; white arrowheads point to a miscounted cell in red rectangular ROIs. (**F**) Comparison between automated Quanty-cFOS and human manual counts over 30 images; blue line shows the automated Fos-positive cell counts; orange, yellow, purple and green lines represent the manual counts performed by the four experimenters. (**G**) Automated Quanty-cFOS counts and average of manually counted Fos-positive neurons; blue circular markers show the automated counts, red circular markers the average values of manual counts. Single manual Fos-positive cell counts performed by each of the four experimenters are shown by the orange, yellow, purple and green triangular markers. The dashed lines show the counting slope for the automated Quanty-cFOS (blue) and the human manual counts (red). (**H**) z-score analysis with a significant counting correlation in the two standard deviations (S.D.) range between the automated and human manual counts. Blue and red circles plot the individual correlation values for the automated and the manual counts. Black dashed line shows the correlation slope.

**Figure 4 cells-12-00704-f004:**
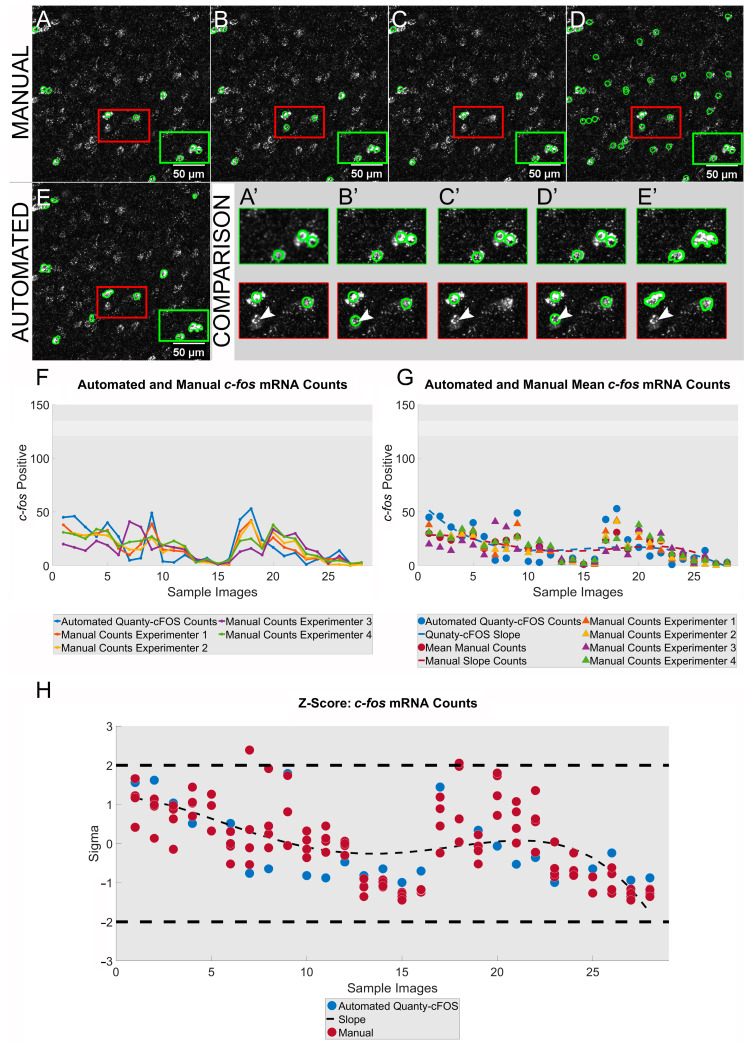
*c-fos* mRNA: analysis and comparison of cells counted manually vs. Quanty-cFOS. (**A**–**D**) Representative maximum intensity projection images for *c-fos*-positive cells manually counted by four experimenters. Scale bar was added using ImageJ/Fiji on the right bottom corner of each image. (**E**) Quanty-cFOS automated *c-fos*-positive cell counting using the ‘*Automated Optimization*’ method. (**A**–**E**) *c-fos*-positive cells are highlighted with a green outline. (**A**–**E**,**A’**–**E’**) Green rectangular ROI shows cells counted manually by the four experimenters to be positive for *c-fos* mRNA or counted as positive using the Quanty-cFOS automated method. Red rectangular ROI shows Fos-positive cells counted differentially between the four experimenters and the Quanty-cFOS tool. (**A’**–**E’**) Highlighted red and green rectangular ROIs; white arrowheads point to a miscounted cell in red rectangular ROIs. (**F**) Comparison between automated Quanty-cFOS and human manual counts over 30 images; blue line shows the automated *c-fos*-positive cell counts; orange, yellow, purple and green lines represent the manual counts performed by the four experimenters. (**G**) Automated Quanty-cFOS counts and average of manually counted *c-fos*-positive neurons; blue circular markers show the automated counts, red circular markers the average values of manual counts. Single manual *c-fos*-positive cell counts performed by each of the four experimenters are shown by the orange, yellow, purple and green triangular markers. The dashed lines show the counting slope for the automated Quanty-cFOS (blue) and the human manual counts (red). (**H**) z-score analysis with a significant counting correlation in the two standard deviations (S.D.) range between the automated and human manual counts. Blue and red circles plot the individual correlation values for the automated and the manual counts. Black dashed line shows the correlation slope.

**Figure 5 cells-12-00704-f005:**
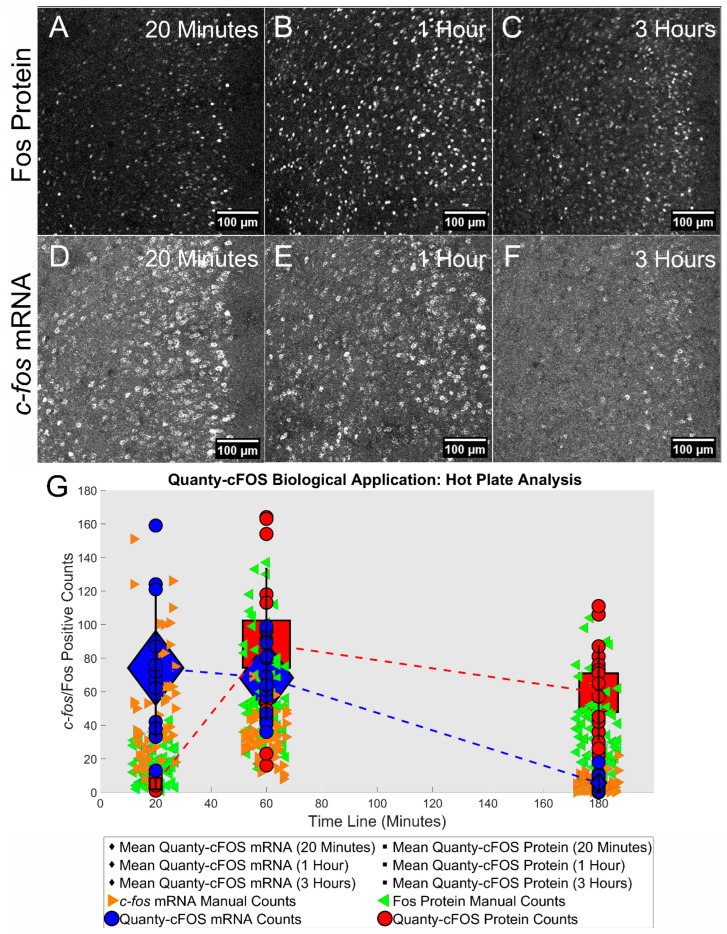
Using Quanty-cFOS to study Fos protein and *c-fos* mRNA expression in the brain over time following sensory stimulation of the hindpaw. Analysis of cells expressing Fos protein (**A**–**C**) and mRNA (**D**–**F**) at three different time points after hindpaw exposure to a 50 °C heat stimulus is shown. Representative images at 20 min (**A**,**D**), 1 h (**B**,**E**) and 3 h (**C**,**F**) after heat stimulation of the paw are shown. The gamma was adjusted to 0.6 in ImageJ/Fiji for the purpose of representation. (**G**) Statistical analysis of mean Fos protein and *c-fos* mRNA expression. Red circles show the Quanty-cFOS protein automated counts on 16 images for each time point; the red rectangular boxes highlight the mean Fos-positive automated counts. The green left arrowheads indicate the protein manual counts performed at the 3 time points. Blue circles show the *c-fos*-positive counts using Quanty-cFOS automated counting on 16 images for each time point; the blue diamond boxes highlight the mean *c-fos*-positive automated counts 20 min, 1 h and 3 h after heat plate stimulation and the orange right arrowheads show the mRNA manual counts. Dash lines highlight the changes in positive cells counts over time. Data are represented as mean ± S.D.

## Data Availability

All raw data are included in the figures in form of individual data points in dot blots. All Matlab scripts and algorithms generated during the development of Quanty-cFOS are uploaded in a publicly accessible repository: https://github.com/cberri/Quanty-cFOS (accessed on 15 February 2023).

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
