# Peer review of "Quanty-cFOS, a Novel ImageJ/Fiji Algorithm for Automated Counting of Immunoreactive Cells in Tissue Sections"

_cells, 2023, doi:10.3390/cells12050704_

Round 1
Reviewer 1 Report
The study by Beretta et al is a well-prepared manuscript, elegantly written, with a very good background report of the state-of-the-art and with a methodology with adequate controls supporting the relevance of the proposed algorithm for automated counting of immunoreactive cells in tissue sections. Very elegant and clear-cut images are provided. The immunocytochemical detection involved the counting of (i) cells with nuclei positive for the Fos protein resulting from the expression of c-Fos proto-oncogene; and (ii) cells with cytoplasm positive for c-fos mRNA on images derived from tissue sections. In the end, I do not think it is an easy-to-use program even with the videos supporting the paper and respective legends. However, that might be difficult of hands-on of the present reviewer. Nothing that the authors could not solve if someone has difficulties and contact them.
I just have a few things to be elucidated and corrected.
Comment #1: No one-to-one practical advantages/disadvantages of the Quantify-cFos are made in comparison with other automated cell counting programs available in the scientific community. Although the authors used several experimenter countings’ results, a comparison group with at least one of these available softwares could be an important point in choosing the present solution.
Comment #2: On page 3 (lines 101-103) the authors refer that “A heat stimulus was presented to mice on a hot plate at 50 ℃ for 30 s”. This seems very aggressive for the animal. The usual cut-off behavior is jumping or licking one of the paws. How did the authors manage to keep the animals in the hot plate for 30 s at 50ºC? How does this keep with ethical guidelines since the authors did not evaluate the threshold at 50ºC but kept the animals regardless of their behavior?
Comment #3: In the Materials and Methods section there are a couple of internet links that were not available when this referee tried to reach them:
Page 5: https://github.com/cberri/Quanty- 217cFOS - lines 217-218.
Page 5: https://www.ilastik.org/documentation/pixelclassification/pix- 233elclassification – lines 233-234
Please check.
Comment #4: Why does the text has so many words that need to be rewritten due to being hyphenated?
Ex:
-Title: “immunoreac-tive”
Abstract “hu-man”, “de-scribe” “pro-cess”, “mak-ing”, “la-belled” and so on along the manuscript.
Please correct these all.
Reviewer 2 Report
The authors describe an open-source ImageJ/Fiji tool, called ‘Quanty-cFOS’, with an easy-to-use, stream-lined pipeline for automated or semi-automated counting of cells positive for Fos protein and/or c-fos mRNA on images derived from tissue sections. The algorithms compute the intensity cutoff for positive cells on a user-specified number of images and apply this on all the images to process. This allows to overcome variations in the data and enables deriving cells counts registered to specific brain areas in a highly time-efficient and reliable manner.
The Quanty-cFOS tool was validated using data from brain sections following sensory stimulation of the hindpaw in a user-interactive manner.
Quanty-cFOS facilitates a rapid, accurate and unbiased spatial mapping of neural activity and can also be easily extended to count other types of labelled cells.
Critique
1. In Figure 1, please show the staining of DAPI to reflect the locations and numbers of nuclei.
2. The wordings are occasionally awkward. E.g., in Lines 550-551, “Analysis of cells expressing Fos protein (A-C) and mRNA (D-F) at three different time points after hindpaw exposure to a 50°C heat stimulus” is not a sentence. Please go through the grammar usage in the manuscript carefully.
3. Can the authors draw a conclusion based on the datapoints shown in Figure 5?
